# Bile canaliculi formation in primary hepatocytes requires α1β1 integrin-dependent adherens junction re-organization

**David Cohen, Francisco Lázaro-Diéguez and Anne Müsch***

## ABSTRACT

Hepatocytes, the parenchymal cells of the liver, exhibit a unique epithelial polarity phenotype in which their bile canaliculi-forming luminal domains and underlying F-actin-linked cell–cell adhesion belt organize not parallel but perpendicular to their basal extracellular matrix (ECM)-contacting domains. Hepatocytes also differ from other epithelia in that they form two basal domains on opposite sites, face only a sparse ECM and express mesenchymal rather than epithelial-typical integrins. What role these hepatocyte-specific cell–ECM interactions play in the establishment and maintenance of the unique hepatocyte polarity phenotype is unknown. We report that in primary rat hepatocyte cultures, development and maintenance of a bile canaliculi network requires the repression of contractile substrate-parallel cell–cell adhesions near matrix-contacting sites. This occurs only when cells contact ECM at two sites; it requires the integrin α1β1, and on rigid matrix, additionally an αV-integrin. We furthermore found that low matrix rigidity, as characteristic of the healthy liver, favors bile canaliculi formation, which becomes independent of p120 catenin-dependent adherens junctions. Our findings thus link the unique hepatocyte polarity phenotype to adherens junction formation downstream of their unique ECM and integrin makeup.

KEY WORDS: Hepatocyte polarity, Bile canaliculi, Junctional crosstalk, α1β1 integrin, αV integrin

## INTRODUCTION

Hepatocytes, the parenchymal epithelial cells of the liver, organize with a unique polarity phenotype that differs from that of other epithelia (Gissen and Arias, 2015; Schulze et al., 2019; Treyer and Musch, 2013). In contrast to monolayered epithelia, which organize their apical domains opposite a basal domain to form ducts or ascini with a central lumen (Fig. 1E, ductal), hepatocytes, organize in single cell cords, flanked by blood vessels on either side of two basal domains, while their luminal domains interrupt cell–cell contacting domains of neighboring cells (Fig. 1E, hepatocytic). The resulting luminal channels connect to form the bile canaliculi (BC) network, through which hepatocytes transport bile acids into a common bile duct. Cirrhosis and hepatocellular carcinoma, end stages of common liver diseases, cause disruption of hepatocyte

Albert Einstein College of Medicine, Department of Developmental and Molecular Biology, Bronx, NY 10461, USA.

*Author for correspondence (anne.muesch@einsteinmed.edu)

D.C., 0009-0008-3110-6189; F.L.-D., 0000-0002-4970-8128; A.M., 0000-0002-7441-7234

polarity ranging from dilation and discontinuities of the BC network to pseudo-ductal organization (Ferraraccio et al., 1997; Goodman, 2007; Kondo and Nakajima, 1987; Meyer et al., 2017; Shousha et al., 2004), leading to backup of bile (i.e. cholestasis), which can in turn exacerbate hepatocyte damage. Yet how the unique polarity phenotype of hepatocytes and their BC network arises and what leads to polarity defects in disease is still poorly understood. Among cell biological features that distinguish hepatocytes from ductal epithelia are the lack of a basement membrane (Amenta and Harrison, 1997; Martinez-Hernandez and Amenta, 1993), the presence of only sparse ECM in the Space of Disse, which separates hepatocytes from the adjacent endothelial cells (Burkel and Low, 1966; Rojkind and Ponce-Noyola, 1982), and hepatocyte expression of epithelial-atypical mesenchymal integrins (Couvelard et al., 1998), mainly α1β1 (Volpes et al., 1991) and α5β1 (Couvelard et al., 1998). Notably, deposition of a basement-membrane like matrix, accompanied by increased matrix rigidity (Lee and Friedman, 2011; Wallace et al., 2008) and changes to hepatocyte integrin expression (Nejjari et al., 2001; Yuan et al., 2000), are hallmarks of fibrotic liver disease, a condition that can lead to hepatocyte polarity defects. Given the instructive role of ECM–integrin signaling established for morphogenesis of ductal epithelial cells (Lee and Streuli, 2014; Matlin et al., 2003; O'Brien et al., 2001; Ojakian and Schwimmer, 1994; Wang et al., 1990), the unique ECM and integrin makeup of hepatocytes makes cell adhesion signaling a key candidate determinant of their unique polarity phenotype. Such role for ECM is consistent with the established practical knowledge that primary hepatocytes isolated from the liver by collagenase digestion can re-polarize when cultured sandwiched between layers of ECM on opposite sides, but de-differentiate when in contact with ECM at only the plating surface (Dunn et al., 1989, 1991; Yang et al., 2016; Zeigerer et al., 2017). Although the effects of variations in ECM composition on the differentiation status of cultured hepatocytes have been extensively investigated (e.g. Bissell et al., 1986, 1987; Deharde et al., 2016; Ong et al., 2018; Sharma et al., 2010; Watanabe et al., 2016) those efforts were largely focused on optimizing culture conditions for maintenance of differentiation markers, with few interrogations into underlying polarity-stimulating signaling mechanisms (Bebelman et al., 2023; Belicova et al., 2021; Fassett et al., 2006; Godoy et al., 2009; Li et al., 2016). Meanwhile, from studies in hepatoma-derived cell lines, evidence has emerged for a role of cell–cell adhesion mechanisms in the polarity decision between ductal and hepatocytic lumen organization (Braiterman et al., 2008; Konopka et al., 2007; Lazaro-Dieguez and Musch, 2017; Theard et al., 2007). In ductal epithelial cells, where a contractile sub-apical cell–cell adhesion belt and the zonula adherens (ZA) and focal adhesion (FA)-associated contractile F-actin bundles are oriented parallel to each other (Fig. 1E, ductal, double-headed arrow indicates force vector), a coordinated regulation, such that increased traction forces are matched by increased cell–cell adhesion forces (Borghi et al., 2010; Martinez-Rico et al., 2010; Maruthamuthu et al., 2011),

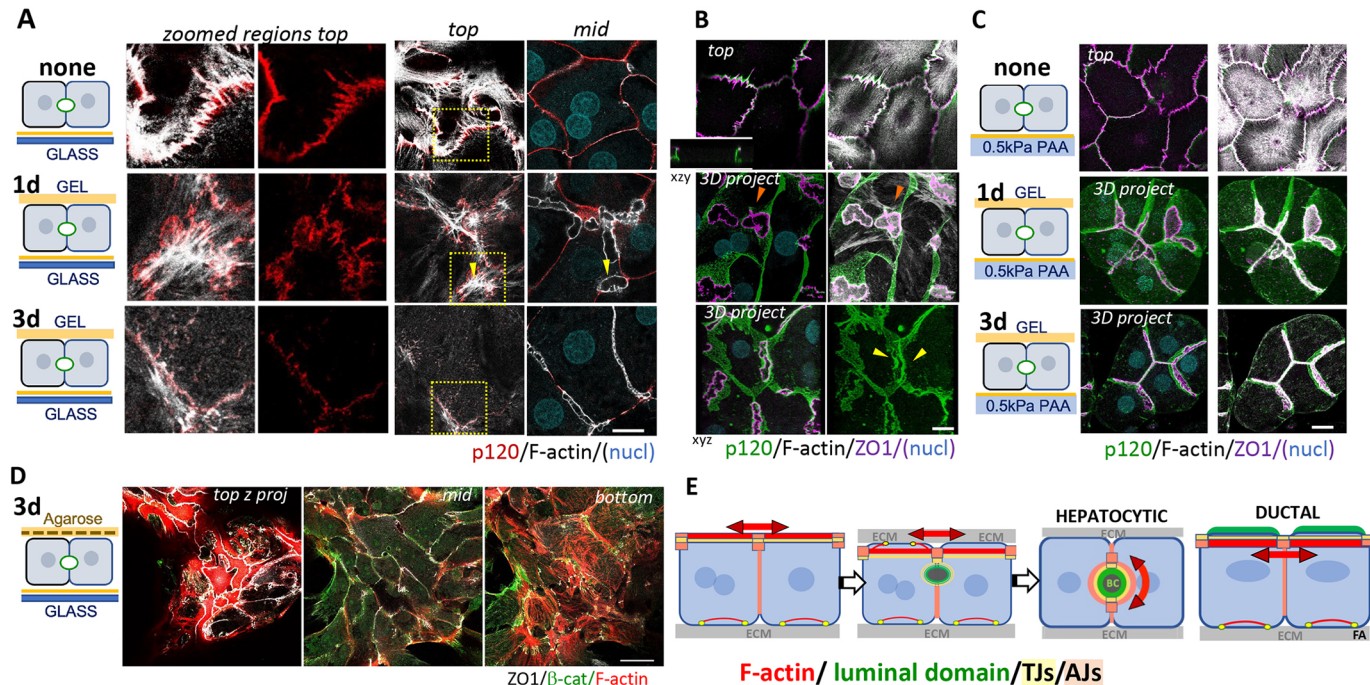

**Fig. 1. BC formation after collagen overlay is accompanied by loss of AJ-associated substrate-parallel stress fibers.** Hepatocytes were plated on either collagen-I-coated glass (A,B,D), or on collagen-I functionalized PAA of 0.5 kPa (C) as indicated in the adjacent cartoons. The top row of A–C shows non-overlaid cells (none), with (A) F-actin stress fibers, linked to the AJ protein p120-catenin (p120), span across the free surface (top) and no BC are apparent (mid), or (B,C) TJ marker ZO1 forming a substrate-parallel circumferential ring at the apex, see also inserted x-z-view in B. The middle row of A–C shows 1 day O/L. At 1 day after collagen-I O/L, ZO1-lined BC lumina form; in cells plated on glass (A and B), AJ-anchored F-actin is still present above the nascent BC (arrowheads in A) and, F-actin filaments, contacting p120-catenin, emanate tangentially from ZO1-lined BC (orange arrowhead in B), whereas top stress fibers are no longer apparent in cultures on 0.5 kPA (C). The bottom row of A–C shows 3 day O/L panels. At 3 days after O/L, F-actin bundles spanning across the top surface have disappeared in all cultures and BC have elongated; p120-catenin is enriched around ZO1-delineated BC (yellow arrowheads in B) while also being present along the length of cell–cell contacts. (D) Merged top, mid and bottom planes of 3-day O/L cultures, when 0.5% agarose was included in the overlay matrix. Note horizontal TJ on top. (E) Scheme of ductal epithelial and hepatocytic lumen polarity and organization of their associated TJ and AJ junctions; red double arrows indicate force vectors on their sub-luminal cell–cell adhesions; also schematically illustrated from left to right are the steps in the establishment of hepatocyte polarity in collagen overlay cultures, as identified in A–C. Scale bars: 20 μm (A,B), 50 μm (C,D). nucl, nucleus (DAPI staining). Note that the bottom sections of the images in A are shown in Fig. 6A. Images shown are representative of three (A–C) or two (D) repeats.

is thought to enable epithelial monolayers to withstand mechanic stress. We hypothesize that hepatocytes that organize their ZA in such a way as to be perpendicular, and not parallel, to their basal domain (Fig. 1E, hepatocytic) are adapted to low matrix rigidity and are less able than ductal epithelia to maintain cell–cell junctions when confronted with tangential tension on rigid ECM. To gain insight into the role of hepatocyte crosstalk between adherens junction (AJ) and integrin adhesions, we investigated how manipulation of substrate rigidity and integrin functions in primary rat hepatocyte cultures affected F-actin organization, and cell–ECM and cell–cell adhesions, as well as BC network integrity. Our analysis revealed that hepatocyte cultures respond to increased rigidity of a collagen matrix with the formation of substrate-parallel AJ-associated F-actin bundles and that, on rigid matrix, integrin α1β1 along with *de novo* assembled αV-integrin-containing adhesions play protective roles in the maintenance of the BC network.

## RESULTS
### A soft overlay matrix causes re-organization of substrate-parallel to substrate-perpendicular cell–cell junctions and dissolution of apical stress fibers to allow BC formation

Primary rat hepatocytes, isolated by liver perfusion with collagenase, have been shown to re-acquire polarity, forming extensive BC networks when cultured in collagen sandwiches (Dunn et al., 1991,

1989; Yang et al., 2016). In this culture modality, the cells are plated on a collagen substrate at a density that allows cell–cell contact formation and are later overlaid with gelling collagen on their apex. We found that upon collagen plating, with a substrate contacting and a free surface, primary rat hepatocytes assemble, like ductal cells, circumferential substrate-parallel AJs and tight junctions (TJs) near the free surface (Fig. 1A–C, upper panels, zoomed regions in A). Notably, apical AJs localized to F-actin fibers, spanning across the free surface (Fig. 1A–C, upper panels). Collagen overlay caused a TJ re-organization from being substrate-parallel to substrate-perpendicular, as nascent F-actin-enriched BC structures formed [Fig. 1E, hepatocytic; Fig. 1A,B, 1 day, 3 days (marked as 1d and 3d in figures) after collagen overlay (O/L)]; AJs concentrated in a ZA around them, while persisting at lower fluorescence intensity along the length of the lateral domains (Fig. 1B, 3 day O/L, yellow arrowheads); AJ-linked stress fibers at the apex above nascent BC were still apparent at 1 day O/L (Fig. 1A, 1 day O/L arrowheads), and can be seen contacting the ZO1 (also known as TJP1)-lined BC (orange arrowhead in Fig. 1B, 1 day O/L), but F-actin bundles spanning across the cell apex disappeared from the overlay surface in fully polarized cells (Fig. 1A, compare 1 day to 3 days). When 0.5% agarose was included into the collagen-I overlay gel, junction re-organization, dissolution of apical F-actin and BC initiation all failed to occur (Fig. 1D). Based on different sources and methods of measurement, rat tail collagen-I at 0.6 mg/ml, which we used as

overlay gel, likely has a Young's modulus of Elasticity, a material property measuring stiffness, at or below 1 kPa (Paszek et al., 2005; Raub et al., 2010), whereas 0.5% agarose is estimated to raise the Young's modulus of the overlay gel by a magnitude or more (Ulrich et al., 2010). Together these data thus indicate that cell–cell junction re-organization accompanying BC formation requires (1) ECM interaction on opposite surfaces and (2) a soft overlay matrix.

To investigate the effect of rigidity of the plating matrix, we maintained the collagen concentration but varied its stiffness by plating cells on either collagen gels similar to the overlay gel, on collagen-I-functionalized polyacrylamide (PAA) gels of estimated ~0.5 kPa or ~20 kPa (Tse and Engler, 2010), or on collagen-I-coated glass (>50 GPa) (Callister and Rethwisch, 2022) (Fig. 1B versus 1C; Fig. 2A). We determined that re-organization of AJs and TJs from matrix parallel-to-perpendicular upon collagen overlay occurred regardless of plating matrix rigidity and that all matrices were permissible for BC network development (compare Fig. 1B,C). Yet, compared to rigid matrix, plating matrix of low rigidity accelerated BC formation (Fig. 2B, compare glass versus gel and 0.5 kPa versus 20 kPa at 1 day and 3 day O/L), and yielded a more complete maximal BC network (Fig. 2A,B, 3 day O/L), which we quantified as the BC length fraction, the fraction of cell–cell-contacting surfaces associated with BC. Lower BC length fractions in cultures on 20 kPa and glass correlated with stress fiber formation at the rigid domain (Fig. 2A, bottom panels), and increased cell spreading when compared to cultures on 0.5 kPa PAA (Fig. 2C). Thus, in hepatocyte

culture, low matrix rigidity and lack of stress fibers is associated with faster and more extensive BC formation.

### α1β1 integrin, in a matrix-rigidity-dependent manner, prevents substrate-parallel AJs on the overlay domain

α1β1 integrin represents the only collagen-binding integrin in hepatocytes (Couvelard et al., 1998; Volpes et al., 1991). As such, we expected α1β1 to mediate the detrimental effects of collagen rigidity on hepatocyte polarization and anticipated that inhibiting α1β1 after cell attachment and collagen overlay would improve polarization of hepatocytes cultured on rigid collagen matrix. Instead, we observed the opposite – α1β1 inhibition, with a function-blocking antibody (clone 3A3; Turner et al., 1989) (Fig. 3B,C,E) or with the disintegrin peptide Obtustatin (Marcinkiewicz et al., 2003) (Fig. 3D,F–J), as well as α1 (ITGA1)-integrin depletion by siRNA (Fig. 3E; Fig. S1) during 3 days of culture upon collagen overlay all reduced the BC length fraction (Fig. 3E,I, glass; Fig. S1B,C, glass). The BC phenotype was associated with formation of myosin II-positive F-actin bundles (Fig. 3D) linked to AJs spanning the top of the cells (Fig. 3C,F), similar to the F-actin organization observed in control cultures during polarity development transiently (compare to Fig. 1A,B, 1 day O/L). As in that scenario, the AJs linked to horizontal F-actin bundles in α1β1-inhibited cells localized above remnant BC (Fig. 3F); indeed, F-actin connected BC remnants, identified by their ZO1 labeling, to the cell–cell junctions on top (Fig. 3C, arrow and asterisk). We surmise that this stress fiber linkage causes tension on cell–cell

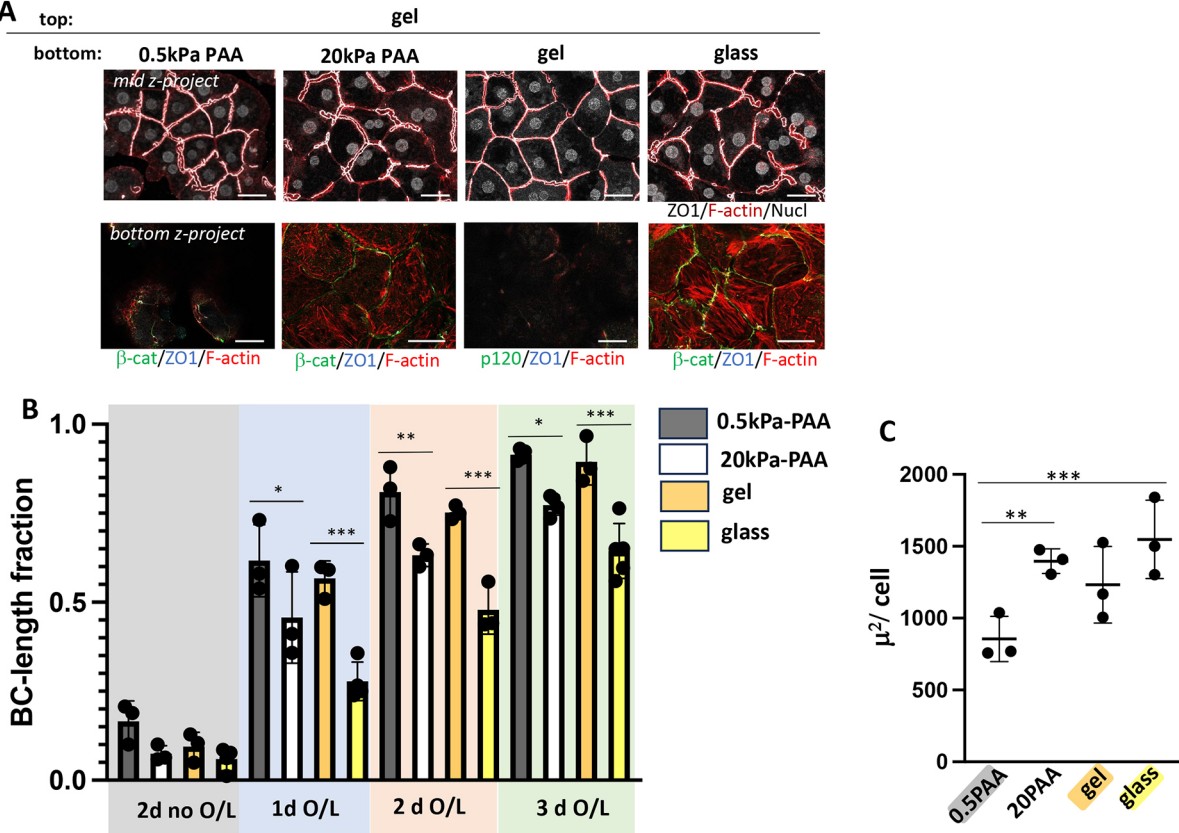

**Fig. 2. BC formation and elongation benefits from low matrix rigidity.** (A) Merged mid and bottom planes (z-projection) of hepatocyte cultures plated either on collagen-I gel, on collagen-functionalized PAA gels of ~0.5 kPa, or ~20 kPa or on collagen-coated glass and overlaid with collagen-I gel and cultured for 3 days; ZO1 outlines BC, which are also enriched in F-actin. β-cat, β-catenin. Note that rigidity-dependent stress fiber formation at the plating surface was accompanied by reduced BC length fraction. (B) Quantification of BC length fraction for each condition. BC length fractions were compared between glass-gel and 0.5–20 kPa cultures at day 1, 2 and 3 upon O/L. (C) Average cell area of 3 day O/L cultures on the indicated matrices; note that only significant differences are indicated; comparisons between all other datasets were not significant. Data shown are mean±s.d. n=3 or 4 (B); n=3 (C). ***P<0.001; **P<0.01; *P<0.05 (one-way ANOVA with multiple comparisons). Scale bars: 50 µm.

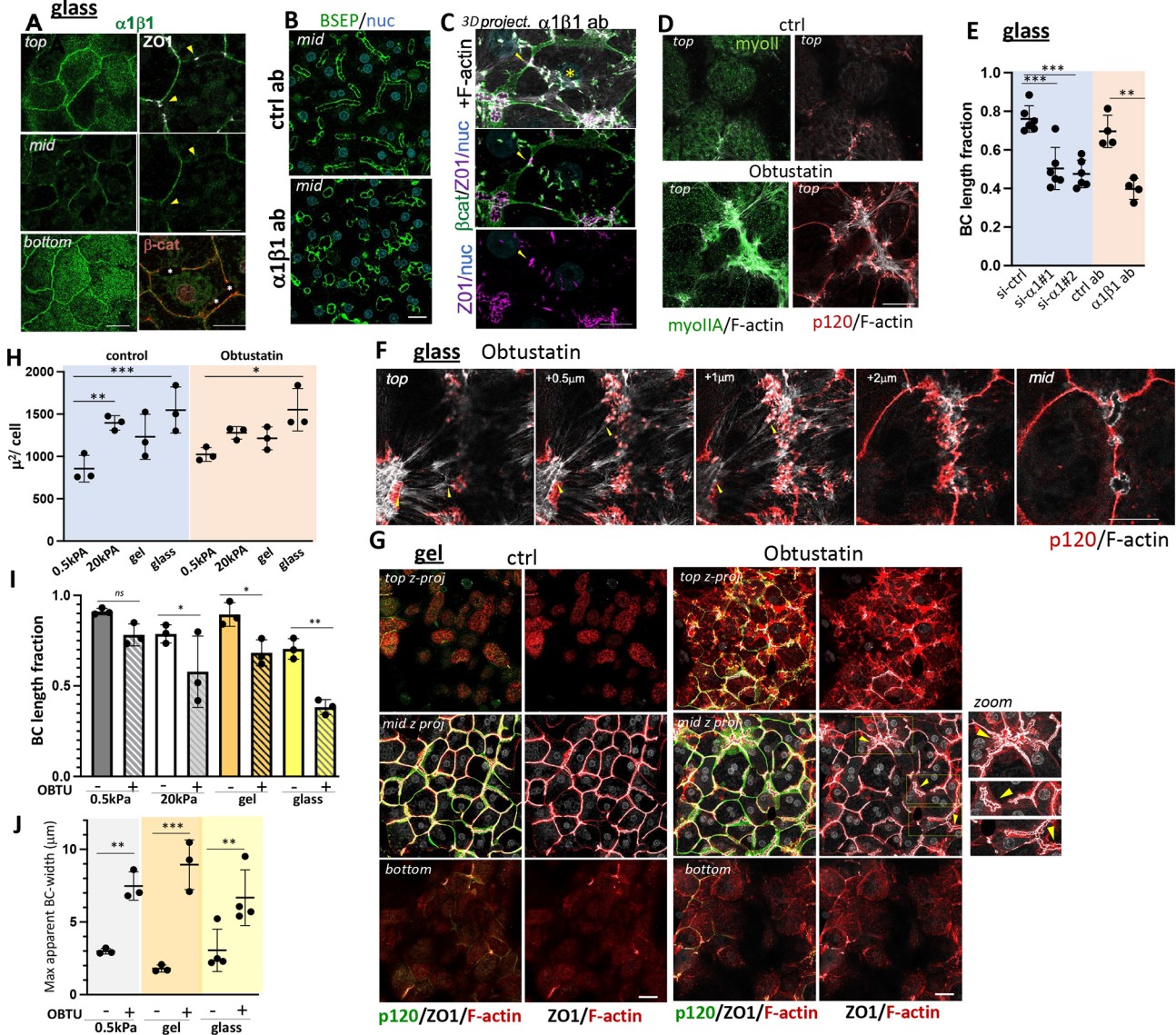

**Fig. 3. α1β1 integrin localizes to AJs and restricts substrate-parallel stress fibers while promoting BC formation.** (A) α1β1 integrin is enriched along the lateral domains and colocalizes with β-catenin, but not with ZO1 (arrowheads). (B,C) Function-blocking α1β1, but not isotype-matched control (ctrl) antibodies, disrupt BC, labeled in (B) for luminal BSEP, when added during 3 days of collagen O/L to cultures plated on glass; projections (C) show ZO1-positive remnant BC (arrowhead) connected to top AJs; note that ZO1 colocalizes with β-catenin on top (asterisk). (D) Top planes of 3 day O/L cultures on glass show that F-actin linked to p120-catenin (p120) in Obtustatin-treated cells is enriched in myosin II. (E) BC length fractions were determined in cultures transfected with ctrl- or α1-siRNAs (si#1, si#2), plated on collagen-I-coated glass and with collagen-I O/L for 3 days, or treated with isotype-matched control IgG or function-blocking α1β1 antibody during the 3 day O/L culture. n=4 (antibody treatment), n=6 (siRNA treatment). (F) Serial x-y-z sections from top to mid image planes of 3 day collagen O/L cultures plated on glass and treated with Obtustatin, showing F-actin linking p120-catenin from across cells (arrowheads); note AJs are above BC remnants seen in mid planes. (G) z-projections of top, mid and bottom planes of control and Obtustatin-treated cultures in collagen-gel sandwiches at 3 day O/L; arrowhead points to dilated BC, enlarged on the right. (H) Average cell area of 3 day O/L cultures on the indicated matrices and cultured in the presence of Obtustatin or under control conditions, as were presented separately in Fig. 2C. n=3. (I) BC length fractions in overlay cultures plated on either collagen-coated glass or collagen gel or on 0.5 kPa versus 20 kPa collagen hydrogels and treated with Obtustatin during 3 days of O/L culture; BC length fractions were compared between glass-gel and 0.5–20 kPa cultures. n=3. (J) Widest distance across BC as it was apparent in merged z-projections in control or Obtustatin-treated overlay cultures on the indicated substrates. n=3 (0.5 kPa, gel), n=4 (glass). Scale bars: 20 μm (A,C,D,F), 50 μm (B,G). Data shown are mean±s.d. ***P<0.001; **P<0.01; *P<0.05; ns, not significant (one-way ANOVA with multiple comparisons). Note that merged top sections of Fig. 3F (top to +1 μm), are also shown in Fig. 6B.

junctions to exceed their adhesion force, causing the BC to fragment or fail to elongate. The latter was apparent when control and Obtustatin-treated cells were observed upon collagen overlay by brightfield timelapse microscopy (Fig. 4A; Movie 1). Under control conditions, BCs arose from the fusion of small separate lumina, which subsequently elongated and became stabilized by previously described 'bulkheads' (Bebelman et al., 2023). In Obtustatin-treated cells, the initial lumina maintained a rosette-like shape and eventually collapsed without elongating.

Besides its effect on rigid cultures plated on glass, as well as on 20 kPa PAA gels [Fig. 3I, 20 kPa for Obtustatin; Fig. S1C, 20 kPa for α1 (ITGA1) RNAi], α1β1 inhibition also caused AJ-associated F-actin bundles on the overlay domain and reduced the BC length fraction in cultures on collagen gels (Fig. 3G,I collagen gel for

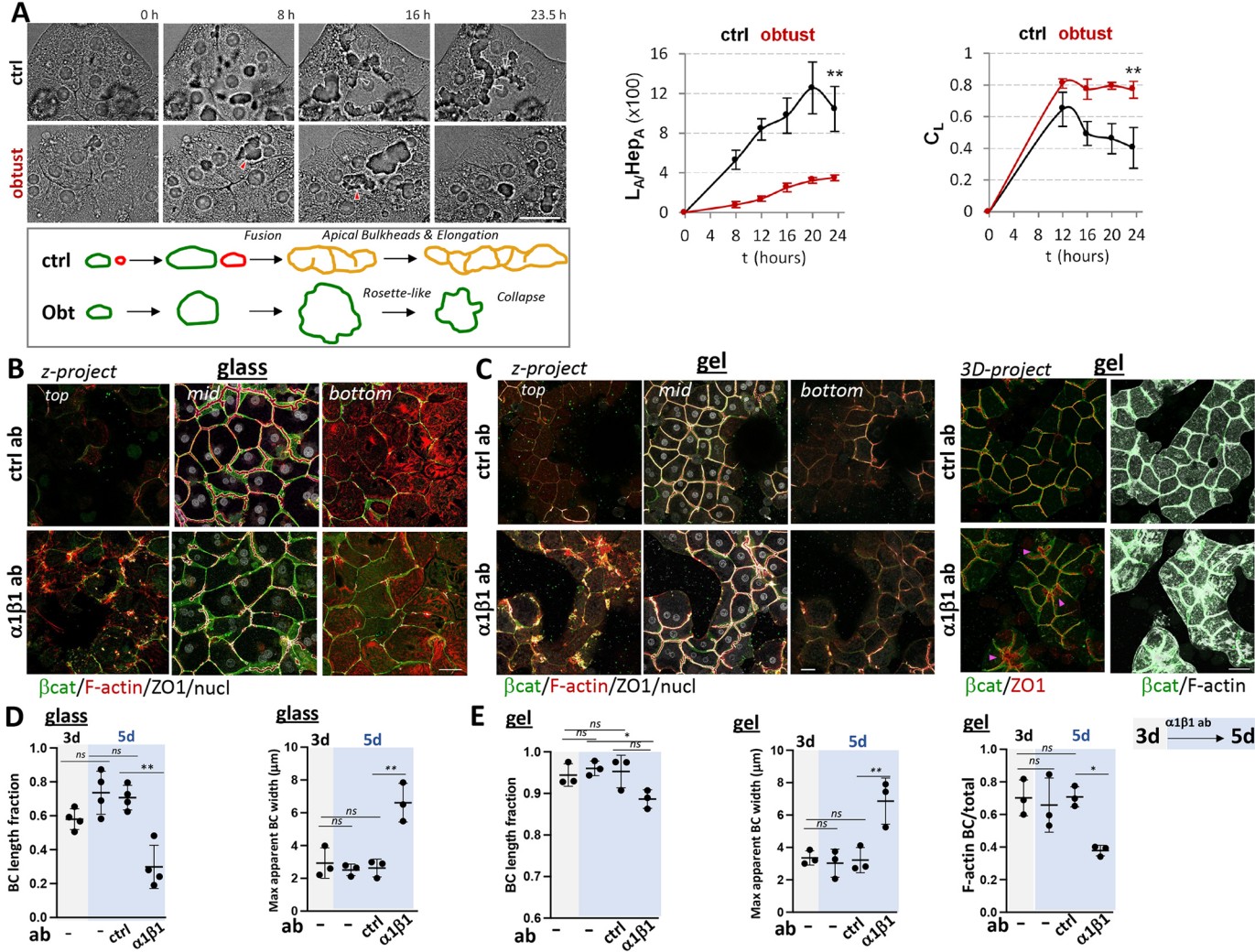

**Fig. 4. α1β1 inhibition interferes with acquisition as well as maintenance of the BC network.** (A) Brightfield timelapse of control (ctrl) or Obtustatin (obtust)-treated cells plated on glass during the first day upon collagen overlay. Gray and red arrowheads indicate lumina with apical bulkheads and rosette-shaped lumina, respectively, as schematically depicted below. Right, ratio of lumen area ($L_A$) to cell area ($Hep_A$) in the image field. $C_L$, circularity of lumina. Total cells analyzed from five fields of an area of ~2.4×10⁴ μm²: control (84), +Obtustatin (75). (B,C) $xyz$-projections of top, mid and bottom planes or 3D cell volume projections (C, right image panel) of cultures 5 days after collagen O/L. They were plated either on collagen-coated glass (B) or on collagen gel (C) and treated at day 3 after O/L (3 days) for 2 days with either ctrl antibody or function-blocking α1β1 antibody (5 days). Arrowheads in the α1β1 panel in C point to dilated BC. (D,E) BC length fraction and maximal apparent BC width quantifications at days 3 (gray) and 5 (blue) of the culture regimen for collagen-glass- (D) and collagen-gel- (E) plated cultures. 'F-actin BC/total' in E refers to the ratio of F-actin area fraction associated with BC to F-actin area fraction in projections spanning the entire cell volume. Data shown are mean±s.e.m. (A; $n$=5 image fields in one experiment) or means±s.d. (D,E; $n$=3, except $n$=4 for BC length in D). **$P$≤0.01; ns, not significant [Mann–Whitney unpaired two-tailed $t$-test applied at time point 23.5 h (A) and by one-way ANOVA with multiple comparisons (D,E)]. Scale bars: 50 μm.

Obtustatin), where stress fiber formation at the plating matrix was not apparent (see Fig. 2A, lower panels); a similar, albeit not significant trend was seen in cultures on 0.5 kPa (Fig. 3I, 0.5 kPa for control versus Obtustatin). Apart from stunting BC length, α1β1 inhibition also distorted the BC, such that their $x$-$y$ coordinates on the bottom were offset from those at the top. In merged $x$-$y$-$z$ confocal images these distortions manifested as BC dilations (Fig. 3G, arrowheads in Obtustatin mid panels and zoomed insets to the right). To quantify this phenotype, we measured the five widest distances across BC per image field in merged $xyz$ views (Fig. 3J). Increased distortions, likely due to uneven top and bottom forces on BC, were apparent in α1β1-inhibited cells on collagen gel and on 0.5 kPa PAA even though the reductions in BC length fraction in the latter cultures were modest (Fig. 3J, compare control versus Obtustatin in 0.5 kPa and collagen gel).

Together, the findings suggest that α1β1 inhibition impairs BC elongation and causes BC distortions by preventing the loss of substrate-parallel contractile cell–cell adhesions from the overlay surface. The data also indicate that the requirement for α1β1 in this process increases with matrix rigidity.

α1β1 itself localized along the entire lateral membrane, excluding the TJs (Fig. 3A), as confirmed in α1-depleted cultures (Fig. S1A,B), but showed only diffuse, and no apparent FA, staining at either basal surface (plating or overlay domain); the inhibition also did not affect cell spreading (Fig. 3H, compare control and Obtustatin panels). This raises the possibility that the integrin might regulate AJs directly in its proximity. Notably, α1β1 inhibition did not alter the activities of the ERK1/2 or p38 family MAP kinases or that of AKT family proteins (Fig. S2), pathways regulated by α1β1 in other cell types (Abair et al., 2008; Chen et al., 2010; Pozzi et al., 1998; Wary et al., 1996),

suggesting these signaling pathways are not responsible for the role of α1β1 in hepatocyte polarity.

To determine whether, in our collagen overlay model, α1β1 participates not only in the establishment of polarity but also in its maintenance, we incubated cultures with α1β1 or control antibody after BC formation was complete (after 3 days of overlay for 2 days). We determined that in polarized cultures on collagen-coated glass, the α1β1 antibody disrupted the already established BC network and increased F-actin and AJ markers on the top surface (Fig. 4B), resulting in a reduced BC length fraction and in distorted BC, reflected by increased maximal apparent BC width (Fig. 4D). When added to polarized cultures on collagen gels, the α1β1 antibody also caused BC distortions, seen in increased maximal BC width (Fig. 4C, arrowheads in right panel and Fig. 4E) and caused increased F-actin across the overlay domain, as apparent in the z-projections of top planes (Fig. 4C, left panels) and in the 3D projection (Fig. 4C, right panels); the appearance of F-actin outside the BC manifested as a reduction in the area fraction of BC-associated F-actin as quantified in Fig. 4E (right panel). There was no significant reduction in the BC length fraction, however (Fig. 4E, left panel). Thus, α1β1 inhibition also affected already polarized cultures and, as observed during the development of polarity, α1β1 inhibition was more deleterious to cultures on rigid matrix than on soft matrix.

It was unexpected, however, that even in collagen sandwiches, where plating and overlay matrix are equal, α1β1 inhibition increased F-actin and cell–cell adhesion proteins selectively at the overlay domain (compare top and bottom z-projections in Fig. 4C left panels). Employing secondary antibody labeling after fixation and permeabilization, we ruled out that the function-blocking α1β1 antibody lacked access to the bottom domain (data not shown). But when we labeled fixed and detergent-permeabilized cells cultured on collagen gels with α1β1 antibody, we observed higher α1β1 fluorescence intensity near the overlay domain than near the plating domain (Fig. S3). We speculate that this α1β1 asymmetry indicates a functional bias towards the top domain. How it comes about is currently unclear. Cells could still sense the underlying coverslip through long collagen fibers (Mullen et al., 2015; Tusan et al., 2018) and/or they could re-model or degrade collagen

(Evans and Barocas, 2009) at culture medium- and substrate-facing domains differently.

All together, the analysis of α1β1 integrin establishes that α1β1 is required for proper BC formation and maintenance, likely by restricting the AJ tension, which is generated by their linkage to substrate-parallel stress fibers in the absence of α1β1.

## BC network formation is dependent on p120-catenin and JAM-A in cultures on rigid but not on soft collagen

BC, like all epithelial luminal domains, are maintained by the integrity of TJs. The rigidity-dependent displacement of TJ along with AJ proteins to the cell apex revealed by α1β1 inhibition, raised the question of whether AJs are obligate for hepatocyte TJ and BC-formation. To investigate this, we siRNA-depleted p120-catenin (encoded by *CTNND1*) (Fig. S4A, p120), which stabilizes the two cadherins, E- and N-cadherin, in hepatocytes (Doi et al., 2007; Gerber et al., 2022) at the cell surface (Davis et al., 2003). As expected, p120-catenin depletion resulted in loss of the cadherin cytoplasmic plaque proteins α- and β-catenin from cell–cell adhesion domains (Fig. S4A, α-/β-catenin). Remarkably, this had little effect on BC formation in cells cultured on 0.5 kPa collagen, but inhibited BC in cultures on 20 kPa and on glass (Fig. 5A,B, control versus p120). Thus, functional AJs become essential for BC formation only when matrix rigidity exceeds the estimates for the healthy liver (<1 kPa) (Ojha et al., 2022). This is consistent with findings that liver-specific p120-knockout in a mouse model had no effect on hepatocyte differentiation or homeostasis (van Hengel et al., 2016).

We predict that reduced AJ tension on soft matrix also accounts for the diminished effects of α1β1 depletion on BC length on soft plating matrices. Although we detected no rigidity-dependent differences in the mean intensity of Vinculin at the ZA (Fig. S5A, control; Fig. S5B), suggesting that α-catenin is in its extended, tensile conformation under all culture conditions (Yonemura et al., 2010; Choi et al., 2012; le Duc et al., 2010; Yao et al., 2014), direct tension measurements on cadherins remain to be conducted.

The requirement for AJs to support TJs and BC on rigid matrix might arise from a rigidity-induced increase in TJ tension, which is

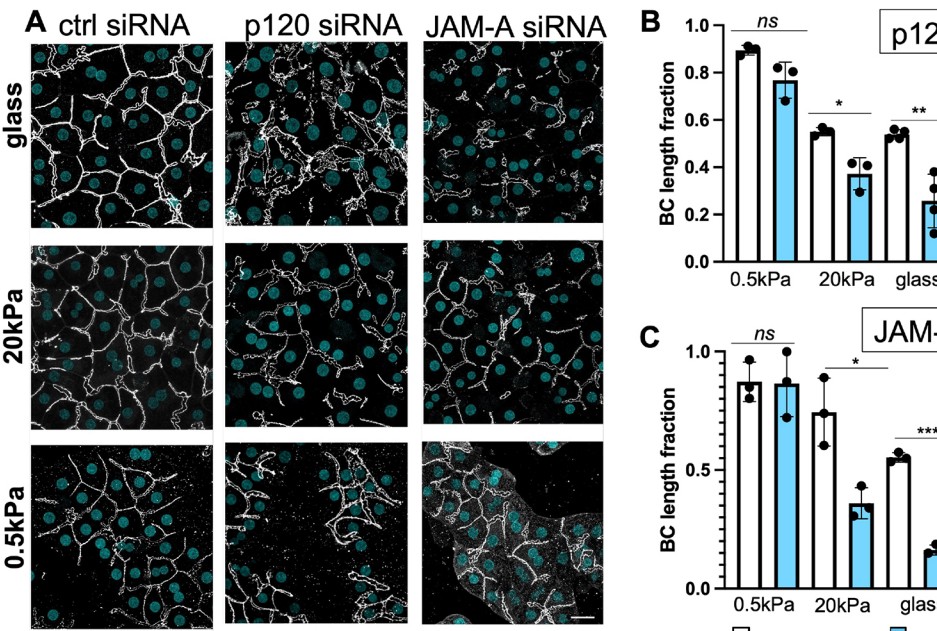

**A** ctrl siRNA | p120 siRNA | JAM-A siRNA

glass / 20kPa / 0.5kPa

*z-project* ZO1/DAPI

**B** p120 — BC length fraction (0.0–1.0) vs 0.5kPa, 20kPa, glass; *ns*, *, **

**C** JAM-A — BC length fraction (0.0–1.0) vs 0.5kPa, 20kPa, glass; *ns*, *, ***

ctrlsi □   p120/JAM-A si ▨

**Fig. 5. Depletion of p120-catenin or of JAM-A inhibits BC network formation in cultures plated on rigid matrix but not in cultures on soft matrix.** (A) z-projections of mid-planes depicting ZO1-labeled BC in control (ctrl) siRNA, p120- or JAM-A-siRNA-treated 3 day O/L cultures plated on either collagen-coated glass or collagen-hydrogels of 20 kPa or 0.5 kPa and their BC length quantifications (B,C). Data shown are means±s.d. (n=3; n=4 for p120 glass). ***P<0.001, **P<0.01; *P<0.05; ns, not significant (one-way ANOVA with multiple comparisons). Scale bar: 50 µm. p120 labeling in the control and p120si images on glass depicted in A are shown in Fig. S4A.

Journal of Cell Science

sensed by ZO1 (Haas et al., 2020). In MDCK cells, a model for ductal cells, a matrix rigidity-dependent increase in ZO1-tension occurred by depletion of the transmembrane adhesion protein JAM-A (also known as F11R) (Haas et al., 2020). We found that, in hepatocytes, siRNA-mediated JAM-A-depletion (Fig. S4B) inhibited the BC network on rigid (20 kPa and glass) but not on soft (0.5 kPa) substrates (Fig. 5A,C, control versus JAM-A), consistent with a matrix rigidity-induced increase in TJ tension forces, which are exacerbated by JAM-A depletion. JAM-A was previously implicated in in the manifestation of hepatocytic polarity in two hepatoma cell lines, where it was required for the formation of cyst-like lumina, resembling nascent BC, between cells (Braiterman et al., 2008; Konopka et al., 2007).

### α1β1 signaling is integrated with that from the fibronectin receptors α5β1 and αV integrin

As α1β1 inhibition exacerbated rather than prevented stress fibers in collagen cultures, we investigated alternative adhesion mechanisms that might participate in rigidity signaling. After ruling out expression of the α1β1-related collagen receptor α2β1 integrin (Fig. S6A,B), we determined that isolated hepatocytes, as previously reported, deposit fibronectin (Stamatoglou et al., 1987; Williams et al., 2011) (Fig. S6C). Upon plating on collagen-coated glass, the fibronectin receptor α5β1 formed strong fibrillar adhesions at matrix-contacting domains, which became less pronounced upon collagen overlay (Fig. 6A). Upon matrix overlay, α5β1 fibrils were also detectable at the top basal domain (Fig. 6B control). This α5β1 population became more intense upon α1β1 inhibition (Fig. 6B, Obtustatin, images and relative normalized mean intensities), and in α1β1-inhibited cells, it was accompanied by formation of fibronectin fibers, which α5β1 assembles when under tension (reviewed in To and Midwood, 2011) (Fig. 6C, control versus Obtustatin). We further observed that a fraction of the substrate-parallel F-actin bundles at the overlay domain in α1β1-inhibited cells coincided with α5β1 fibrils (Fig. 6B, Obtustatin, lower panel, arrowheads), raising the possibility that they contribute to stress fiber formation. We were unable to directly address their role because neither addition of RGD and

ATN161 peptides, which compete with fibronectin for α5β1 interaction, nor a 50% reduction of α5 by siRNA noticeably reduced fibrillar α5β1 adhesions on the top (not shown).

Colabeling for either talin (herein referring generically to talin 1 and 2) or vinculin with p120-catenin revealed, however, that talin, a marker for matrix adhesions, localized in fibrillar structures adjacent to or overlapping with the AJ marker (Fig. 6E, Obtustatin), whereas vinculin, a proxy for tension at cell–matrix adhesions (Dumbauld et al., 2013) as well as at AJs (Choi et al., 2012; le Duc et al., 2010; Yao et al., 2014), colocalized with p120-catenin (Fig. 6D, Obtustatin). This indicates that AJs, but not the fibrillar matrix adhesions, are under tension, which is likely generated by association of AJs with substrate-parallel stress fibers. In control cells, vinculin labeling was absent from the top surface (Fig. 6D, control), even though vinculin equally labeled the ZA in control and α1β1-inhibited cells (Fig. S5, control versus Obtustatin).

We furthermore determined that where hepatocytes contact rigid matrix (glass, Fig. 7A,C, but not on collagen gel or 0.5 kPa PAA, Fig. S6D), they also established matrix adhesions involving an αV-integrin (Fig. 7A, bottom and zoom in both control and α1β1 ab). Pharmacological inhibition of αV with the cyclic RGD peptide commercially known as Cilengitide (Dechantsreiter et al., 1999) (Fig. 7B-G, cycRGD), but not DMSO solvent (Fig. 7B,C,G) or a scrambled control peptide (Fig. 7F,G), caused hepatocytes to lose discrete αV- (Fig. 7C) and vinculin- (Fig. 7D) positive FAs and instead establish contractile F-actin bundles associated with vinculin-positive cell–cell adhesions, (Fig. 7B–D), which also recruit α5β1 fibrils (Fig. 7B); these stress fibers resembled those at the overlay domain in α1β1-inhibited cells, and they were also associated with BC network disruption (Fig. 7F,G). Inhibition of α1β1 together with αV, caused contractile cell–cell adhesions across both basal surfaces (Fig. 7E). Taken together, these findings indicate that although rigidity of the plating matrix dictates the α1β1 requirement for BC organization, it is an αV integrin, which primarily restricts AJ-associated F-actin bundles at the rigid substrate, whereas α1β1 is crucial only at the overlay domain, where αV is absent. A summary of the rigidity-dependent localization and roles of the investigated

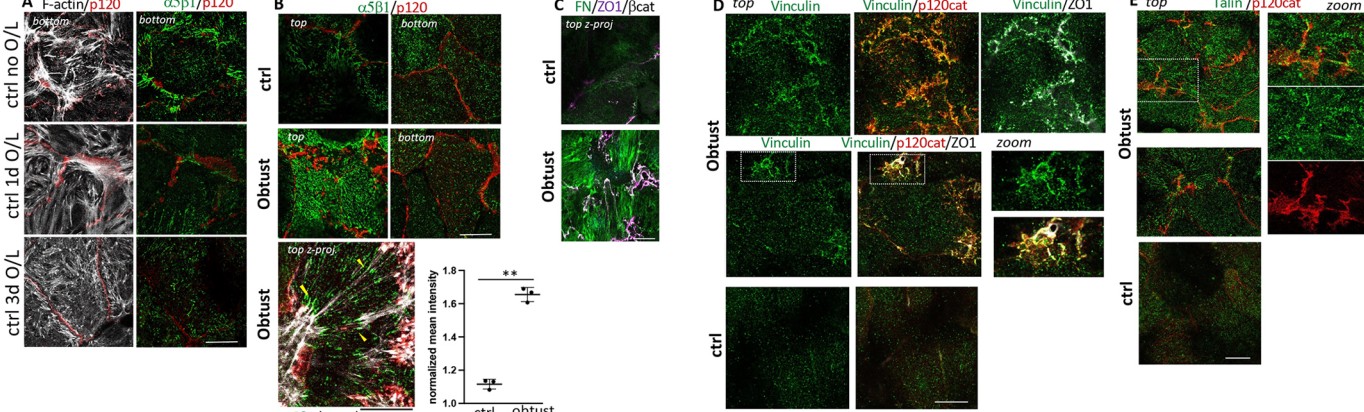

**Fig. 6. Obtustatin induces separate α5β1-positive fibrillar adhesions and vinculin-positive p120-catenin adhesions at the O/L domain.** (A) Bottom planes of the control cells depicted in Fig. 1A showing α5β1 fibrils prior to O/L, which decreased at 1 day and 3 days after O/L. (B,C) α5β1 (B) or fibronectin (FN) (C) labeling with the indicated junctional proteins at O/L domains (top) in control (ctrl) and Obtustatin (obtust)-treated 3 day O/L cultures plated on collagen-coated glass. In B, the bottom planes are also shown, and the top z-projected image of an Obtustatin treated cell (from Fig. 3F) shows α5β1 fibrils adjacent to cell-spanning F-actin fibers (arrowheads). Graph in B shows α5β1 mean intensity at O/L domain, normalized for each experiment to the lowest intensity recorded in the image set. Data shown are means±s.d. (n=3). **P<0.01 (unpaired two-tailed t-test). (D) p120-positive AJs, which overlap with ZO1 at O/L domain (top) of Obtustatin-treated cells, colocalize with vinculin; (E) the p120-positive AJs at the O/L domain (top) are adjacent to or overlapping but not colocalized with talin-positive adhesions. Zoomed images in D,E correspond to the boxed regions in the images to the left. Scale bars: 20 μm. Images shown are representative of three (A–D) or two (E) repeats.

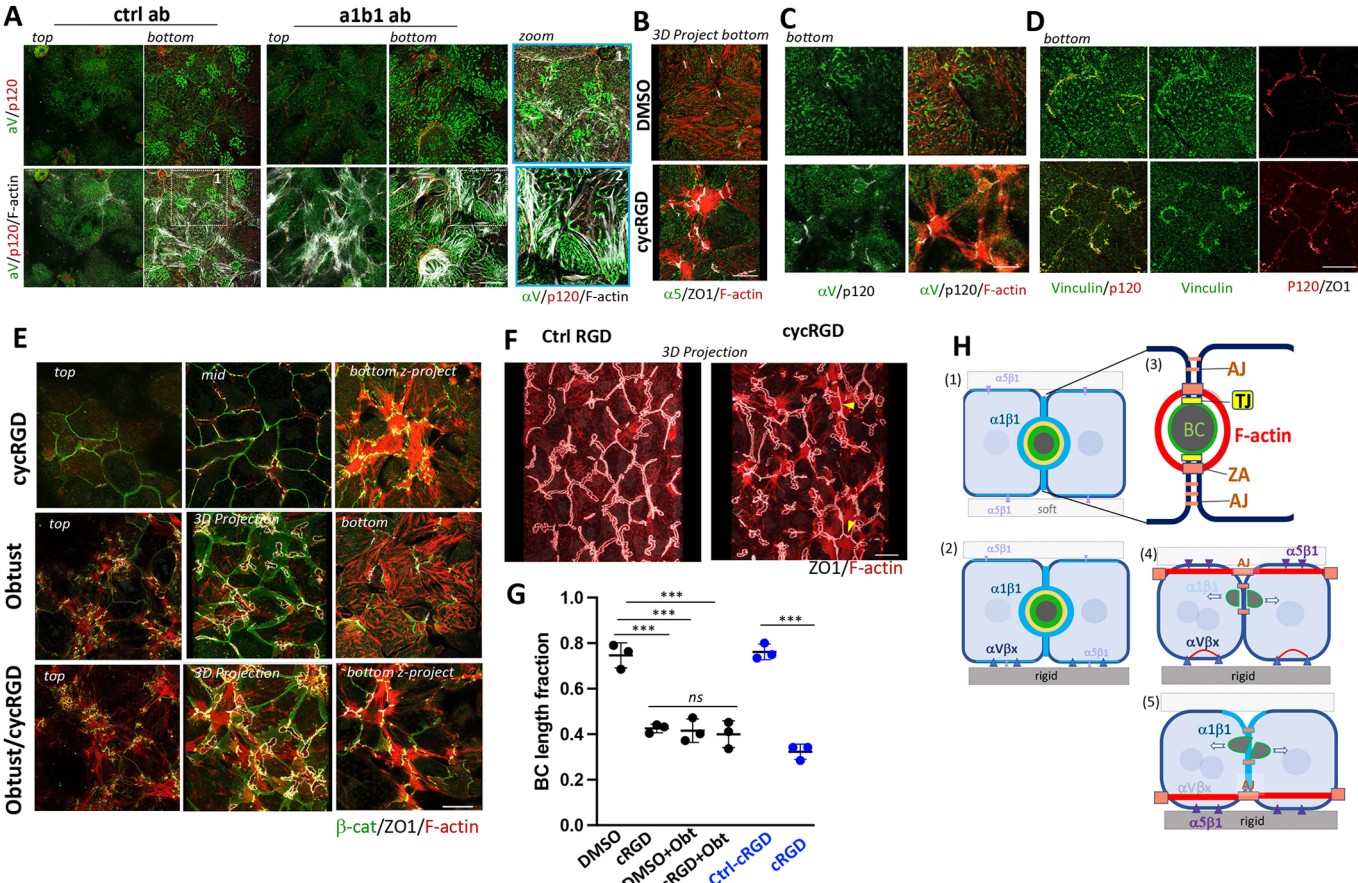

**Fig. 7. α1β1 and an αV integrin independently regulate AJ-linked substrate-parallel F-actin bundles at O/L domain and rigid plating surface, respectively.** Views of top and bottom basal domains (A–E) or mid-planes (E) and 3D projections (B,E,F) of cultures plated on collagen-coated glass, overlaid with collagen and cultured for 3 days in the presence of control (ctrl) or function-blocking α1β1 antibody (A), in the presence of cycRGD or Cilengitide (B–F) or of DMSO solvent (B–D) or a cycRGD control peptide (Ctrl RGD; F), and/or of Obtustatin (Obtust; E). (A) αV forms fibrillar adhesions, which occasionally coincide with F-actin fibers, at the plating (bottom and zoom) but not at the O/L matrix (top) in ctrl and α1β1-inhibited cells. (B) Projections of the bottom planes of cycRGD-treated cells show α5β1 accumulating opposite of ZO1 at the ends of stellate F-actin bundles, which are not apparent under control conditions (DMSO). (C,D) Fibrillar aV (C) and vinculin (D) labeling at attachment planes disappear upon cycRGD-treatment; vinculin instead accumulates at p120-positive AJs, which anchor the stellate stress fibers near cell attachment sites. (E) CycRGD and Obtustatin induce β-catenin-linked stress fibers at bottom and top basal domains, respectively, and on both basal domains when the drugs are combined. Mid-planes and projections show BC disruptions. (F) 3D projections of cultures treated with a cyclRGD control peptide versus cycRGD or Cilengitide; arrowheads in cycRGD point to areas with substrate-parallel circumferential ZO1 organization near the cell attachment domain, resembling the 'chickenwire' TJ organization of ductal cells. (G) Quantified of BC length fraction for results as in E and F. Data shown are means±s.d. ($n$=3). ***$P$<0.001; ns, not significant (one-way ANOVA with multiple comparisons). Scale bars: 20 µm. (H) Schematic summary of rigidity-dependent distributions of integrins and AJs. (1) α1β1, regardless of rigidity, localizes primarily to cell–cell contacts; α5β1 forms small fibrils at both matrix contact sites, while (2) αVβx forms fibrillar adhesions only on glass (rigid). (3) AJs cluster in a ZA linked to a circumferential actin belt around BC and distribute at lower density along the entire lateral domain. (4) Upon loss or inhibition of α1β1, AJs form F-actin linked substrate-parallel adhesions at the overlay domain, which also recruits α5β1 fibrillar adhesions; this phenomenon is most pronounced on rigid matrix and causes disruption of BC. (5) Upon αV-inhibition in cultures on glass, F-actin bundles linked to AJs and α5β1 recruitment occur at the rigid plating matrix and also cause BC disruptions.

integrins for adhesion junction organization and BC integrity is presented in Fig. 7H.

## DISCUSSION

Our studies yielded four principal insights: (1) that hepatocyte polarity requires the repression of contractile, substrate-parallel cell–cell adhesions near matrix-contacting sites, an organization which antagonizes the formation and maintenance of the BC-network; (2) that repression or disassembly of such horizontal junctions occurs only when hepatocytes are in contact with ECM on two sides, as is characteristic *in vivo*; (3) that disassembly and repression of horizontal cell–cell adhesions is favored by low matrix rigidity and requires the integrin α1β1 and, at high matrix rigidity, an αV-integrin; and (4) that increasing matrix rigidity makes hepatocyte BC network

integrity more dependent on p120-catenin and hence, on p120-catenin-dependent cadherin-based AJs, and on JAM-A-mediated suppression of TJ tension.

We thus provide a cell structural explanation for the long-known requirement of a sandwich configuration for re-polarizing isolated hepatocytes in culture and for the benefits a soft plating matrix for BC development. Our data furthermore indicate that the overlay-induced re-organization of cell–cell junctions from 'ductal' to 'hepatocytic' requires the overlay matrix to be soft. We speculate that in reported protocols for primary hepatocyte culture, in which BC formation occurs without matrix overlay but requires helper cells (Bhatia et al., 1998; Cottier et al., 2023; Murakami et al., 2004; Novik et al., 2017), that the latter either enable matrix assembly at the hepatocyte-free surface or provide signals similar to those

Journal of Cell Science

known to be triggered upon collagen overlay. These reportedly include downregulation of ERK1/2, MAPK p38 and AKT proteins, which are kinases associated with epithelial-to-mesenchymal transition (Godoy et al., 2009). Intriguingly, α1β1, which is primarily a mesenchymal integrin, is known to activate these pathways (Abair et al., 2008; Pozzi et al., 1998), but α1β1 inhibition did not affect their activities in our cultures. We consider it possible that α1β1 instead promotes hepatocyte polarization by directly regulating the contractility of AJs, with which the integrin colocalizes along the lateral domain. This is plausible because α1β1 can recruit tyrosine kinases such as FAK and Fyn (Wary et al., 1998), as well as tyrosine-phosphatases (PTPN2, PTPN6 and PRL3) (Mattila et al., 2005; Peng et al., 2006; Singh et al., 2022) and because tyrosine phosphorylation is central to adjusting AJ composition and adhesion strength (McLachlan and Yap, 2007). αV-integrin, on the other hand, likely relieves AJ contractility at rigid substrate-contacting sites by generating traction forces at αV-positive matrix adhesions. Thus, whereas matrix rigidity increases cell–cell adhesion tension, we propose that hepatocyte α1β1 and αV integrins protect cells from excessive AJ tension (Fig. 7H). In this model, tension exceeding cell–cell adhesion forces, and resulting in loss of cell–cell adhesion, causes BC disruptions when α1β1 and αV are inhibited. The distortions of established BC networks after adding α1β1 function-blocking antibodies to polarized collagen sandwich cultures likely resulted from ECM signaling asymmetries between the two basal domains, which we suspect arose even though we provided similar collagen gels on top and bottom. This assumption is based on the observed enrichment of α1β1 at the overlay domain, and the selective assembly of F-actin fibers there when α1β1 was inhibited. Our culture model thus cannot isolate effects resulting from top–bottom asymmetry in matrix rigidity from the effects of rigidity per se. It is conceivable, however, that uneven ECM deposition in the Space of Disse during development of fibrosis leads to similar asymmetries in vivo as well. This possibility is supported by the acknowledgement that due to the focal nature of hepatic stellate cell activation, the main source of fibrogenic collagen in the Space of Disse, collagen deposition is highly heterogenous (Kawamura et al., 2022; Ratziu et al., 2005). Recent 3D volumetric imaging of collagen in liver biopsies confirmed a high labeling variability within the volume of ∼200 µm optically cleared liver sections (Yigit et al., 2025), although it still remains to be determined if this variability is present at the level of hepatocyte cords.

The in vivo functions of hepatocyte integrins are still poorly characterized; yet, a dilation and disruption of the BC network consistent with that of α1β1 or αV inhibition in culture, has been reported upon hepatocyte-specific knockout (Masuzaki et al., 2021) and RNAi depletion (Speicher et al., 2014) of β1-integrin, the major β-integrin subunit in hepatocytes, which pairs in hepatocytes with α1, α5 and α9 (Couvelard et al., 1998; Speicher et al., 2014; Yuan et al., 2000) and can also form heterodimers with αV. Because BC defects in the β1-depletion model manifested only weeks after loss of β1-integrin, and given that whole body α1-knockout mice feature increased hepatocyte-mediated collagen deposition in the Space of Disse (in basal conditions and in fibrosis) (Williams et al., 2015), the reported β1-knockdown defects were likely accompanied by increased ECM rigidity, a condition in which our findings predict α1β1 to be crucial for BC network integrity.

Induction of αV-integrins and α1β1 expression in mesenchymal cells contribute to the differentiation of myofibroblasts, which are drivers of tissue fibrosis (Patsenker and Stickel, 2011). Owing to their roles in these nonparenchymal cells, function-blocking integrin antibodies are being tested in anti-fibrosis therapy (Tacke et al., 2023). Our findings suggest that in the liver such strategies

have a potential downside by interfering with the ability of hepatocytes to adapt to increased matrix rigidity.

## MATERIALS AND METHODS
### Cell culture
#### Rat hepatocyte isolation and culture
Primary rat hepatocytes were isolated from 6–8-week-old male Sprague–Dawley rats by two-step collagenase perfusion as described previously (Neufeld, 1997). The animal studies were approved by the Einstein institutional review committee.

$1.5 \times 10^5$ cells/cm$^2$ were plated on either ddH$_2$O-washed 12 mm German Glass coverslips (Electron Microscopy Sciences #72290-04) coated overnight with 10 µg/cm$^2$ rat tail collagen I (Corning, #354236) in cell culture grade water (Corning, #25055CV) or on collagen-coated PAA-hydrogels or on collagen gels; unattached cells were removed 10–15 min after plating and cultures were overlaid 16 h later with 12 µg/cm$^2$ neutralized collagen solution in 1x hepatocyte medium [DMEM (Corning, #17-205-cv), 10% FBS (R&D System, #S11550), 20 µg/ml insulin (Sigma, #I6634), 15 µg/ml hydrocortisone (Sigma #H0888) and 28 ng/ml glucagon (Sigma, #G2044)]. Culture medium was changed daily. Where indicated, 0.5% low melting agarose (Gibco #18300-012) was included into the overlay matrix.

#### MDCK, NRK and HEK293 cells
Cells obtained from Enrique Rodriguez-Boulan, Weill Cornell Medical Collage, NY, USA, were cultured in DMEM with 10% FBS (R&D Systems, #S11550). Cultures used in Fig. S6A,B and Fig. S2 were cultured to confluency. We did not independently authenticate these lines.

#### Fabrication of collagen-functionalized PAA gels and collagen gels
Fabrication was according to the detailed protocol provided previously (Tse and Engler, 2010). Briefly, gels were polymerized on amino-silanated 12 mm circular coverslip(s) by lowering them onto 15 µl drops of a PAA mix placed on chloro-silanated glass slide(s). Gels were prepared from 40% acrylamide (AA) and 2% Bis-acrylamide (Bis-AA) stocks and polymerized with a 1:1000 volume of tetramethylethylenediamine (TEMED) and 1:100 volume of 10% (w/v) ammonium persulfate (APS). For an estimated elasticity modulus of 0.48 kPA the amounts were: 75 µl/ml AA and 30 µl/ml Bis-AA; for gels of ∼19.7 kPa, 200 µl/ml AA and 132 µl/ml Bis-AA were polymerized. Amino-silanation of the coverslips was by sequentially covering them with 0.1 M NaOH, which was heat evaporated, reacting with 3-aminopropyltriethoxysilane (APES) and after extensive rinses, crosslinking with 0.5% glutaraldehyde in PBS. Chloro-silanated glass slide(s) were prepared by spreading them with dichlorodimethylsilane (DCDMS), wiping off the excess coating and rinsing them with ddH$_2$O. Collagen I-functionalization of the PAA gels occurred by UV-crosslinking 0.2 mg/ml sufosuccinimidyl-6-(4′-azido-2′-nitrophenylamino)-hexanoate (sulfo-SANPAH; Pierce Biotechnology) with a UV 360 nm lamp at a distance of 3 inches for 1 h onto the PAA and subsequently reacting the crosslinker with 10 µg/cm$^2$ rat tail collagen I (Corning, #354236) in 50 mM HEPES, pH 8.5 overnight at 37°C. Coverslips were sterilized by UV irradiation at 254 nm in a UV-Stratalinker 2400 prior to use. Collagen gels on coverslips were prepared in a similar manner; instead of the PAA solution, 12 µg collagen/cm$^2$ neutralized in 1× hepatocyte medium was polymerized at 37°C for 30 min and rehydrated in PBS until use.

#### siRNA-mediated protein depletion and antibody treatment
siRNAs were transfected using RNAimax (Invitrogen #13778030) upon cell attachment. siRNAs were as follows: p120-catenin, 5′-UACAGAA-GGUGGUUGUGAUCCUGGGDTDT-3′; JAM-A, 5′-AUCAUAGACGU-GAAGAGAADTDT-3′ (Dharmacon, custom synthesized); ITGA1-si#1 (Sigma, #SASi_Rn01_00102479); ITGA1si #2 (#SASi_Rn01_00102481). Allstars negative control (Qiagen #1027281) was used as RNAi control.

Function-blocking α1β1 antibodies (hybridoma clone 3A3, DSHB) and Myc antibodies (hybridoma clone 9E10) as isotype control were prepared in parallel as tissue-culture supernatants by adapting the hybridoma to serum-free medium (ADCF-Mab, Cytiva #SH30349.01, HyClone);

antibodies were added unpurified as 1:25 dilutions to the culture medium and replenished daily.

## Drug treatment

The following drugs were used: Obtustatin (Medchem Express, MCE #HY-P1408) at 1 µM; Cilengitide (Cayman #22289) at 20 µM; cyclo (-RADfK) RGD negative control (ANASPEC, #AS-62351) at 20 µM; and GRGDSP (Medchem Express MCE #HY-P1740), 20 µM ATN161-TF salt, (Medchem express, MCE HY-13535A) at 10–250 µM.

## Immunofluorescence and microscopy

Cells were fixed in either methanol for 20 min at −20°C (for fibronectin, β-actin, vinculin, JAM-A and BSEP) or with 4% PFA (all other antigens) at 4°C overnight and quenched with 50 mM $NH_4Cl$ in PBS. PFA-fixed samples were permeabilized with 0.3% Triton X-100. Blocking was in 10% FCS, 1% BSA, 10% rat serum (Sigma, #R9759) overnight, as were primary and secondary antibody incubations. Labeling with ZO1 antibody raised in rat was conducted in the absence of rat serum prior to blocking with rat serum and labeling with additional antibodies. The primary antibodies were used in immunofluorescence (IF) and/or immunoblotting (IB) were as follows: p120-catenin (BD Bioscience, clone 98, 1:200 IF), β-catenin (Sigma, #C2206, 1:1000 IF), α-catenin (Sigma, #C2081, 1:1000, IF), JAM-A (BiCell Scientific, #00251, 1:100), vinculin (Proteintech, #26520-1-AP, IF 1:600), ZO1 (clone R26.4C, Stevenson et al., 1986; IF), β-actin [2D4H5] (Proteintech #66009-1-IG, IF: 1:500), MYH9 (Proteintech, #11128-1-AP, 1:250, IF), BSEP (Thermo Fisher Scientific, #PA5-78690, 1:500 IF); talin (Proteintech, #14168-1-AP, 1:250 IF), ITGA5 (BiCell Scientific, #10005, 1:100 IF), αV-integrin (Proteintech, #27096-1-AP, 1:250 IF), ITGA2 (BiCell Scientific #10002, 1:100 IF, 1:500 IB), α1β1 (clone 3A3, DSHB, 1:10; and in house, 1:25), tubulin (clone YL1/2, Novus Biologicals, #NB600, IB 1:5000), AKT (Santa Cruz Biotechnology, sc-8312, 1:500 IB), AKT (S473) (Proteintech, #664441-1-Ig, 1:2000 IB); pThr180/Tyr182-MAPKp38 (Cell signaling, #9211, 1:1000 IB); MAPKp38 (Cell Signaling, #9212, 1:1000 IB); pThr102/Tyr104-ERK1/2 (Cell Signaling, #9101, 1:1000 IB) and ERK1/2 (Cell Signaling, #9102, 1:1000 IB).

DAPI (Sigma #D8417), 0.6 U/ml CF 647–Phalloidin (Biotum, #00041) and the following $F(ab')_2$ fragment secondary antibodies from Jackson Immuno Reseach Laboratory were incubated at a dilution of 1:500 for 1 h at RT: Rhodamine Red™-X (RRX) AffiniPure™ $F(ab')_2$ fragment donkey anti-mouse IgG (H+L) # 715-296-151; Alexa Fluor® 488 AffiniPure™ $F(ab')_2$ fragment donkey anti-rabbit IgG (H+L) # 711-546-152; Alexa Fluor® 647 AffiniPure™ $F(ab')_2$ fragment donkey anti-rat IgG (H+L) # 712-606-153.

Fixed cells were imaged by confocal microscopy on a TCS SP5 confocal microscope (Leica Microsystems, Wetzlar, Germany) using a HCX PL APO 40×/1.25–0.75 oil CS objective or an HCX PL APO 63×/1.4-0.60 oil $λ_{BL}$ CS objective on glass coverslips mounted in nonhardening, glycerol-based aqueous mounting medium (DABCO). Confocal (pinhole, 1 Airy Unit; pixel size, 160.5 nm) x-y-z and x-z-y stacks were taken at a step size of 0.5–1 µm. Images were processed with LAS AF v.2.6.0.7266 (Leica Microsystems), ImageJ (Fiji, version 2.1.0/1.53c) or v.1.52i (National Institutes of Health) and Adobe CS6 (Adobe Inc.) software. Where indicated in the figure legends, a subset of planes was merged using the Z-projection function with maximal intensity; 3D projections were generated with the '3D projection' function, using the brightest point method, with 1 µm slice spacing and 10° angle rotation with interpolation.

Confocal live cell imaging (Fig. 4A; Movie 1) was conducted with the 40× objective. Hepatocytes were plated at ~50×10³ cells/cm² on collagen I-coated four-compartment glass bottom dishes (Greiner Bio-One International GmbH, # 627870) to minimize variability by allowing for parallel comparisons. After 1 h of collagen I overlay, Obtustatin was added (1 µM) to the growth medium in one compartment and the cells were imaged (13 h upon plating) at 37°C in a $CO_2$-enriched environmental chamber. Confocal brightfield images (pixel size, 303.3 nm) were taken using the mark and find function and the resonant scanner of the Leica TCS SP5 microscope system. 24-h x-y-t stacks (1 frame every 30 min) were collected. The movie was created with Adobe Premiere Pro CS6 and ImageJ v1.54p.

## Immunoblotting

Whole-cell lysates in SDS-PAGE sample buffer were subjected to PAGE-electrophoresis; gels were transferred onto Immobilon-FL membrane (Millipore #IPFL00010) and blocked with 5% non-fat milk; primary antibody incubation was in 1% BSA and 1% fish serum or SEA BLOCK (ND-R0999 Novatein Biosciences); DyLight 680- or 800-coupled secondary antibodies were incubated in 5% milk. Blots were imaged with a Laser scanner Fuji Typhoon NIR Plus (Amersham), and analyzed/quantified with LiCor Image Studio Lite software. Where indicated, loading was controlled by tubulin. Uncropped images of western blots from this study are shown in Fig. S7.

## Quantitative analysis of morphology and protein distribution
### Fluorescence intensity measurements

Mean fluorescence intensity in Fig. 6B was determined in individual x-y confocal image planes from auto-thresholded objects using ImageJ. For comparison between experiments, values were normalized to the lowest measured value in a dataset. Vinculin fluorescence intensity at the ZA, denoted M in Fig. S5B, was determined from fluorescence intensity histograms taken along a line perpendicular to the lateral domain near the TJs; M corresponds to the peak intensity at the membrane; the intensities at the valleys of the peak were recorded as soluble vinculin intensity S. Fluorescence intensity histograms taken along a line perpendicular to the lateral domain were also used in Fig. S2A to determine α1β1 and β-catenin intensities 2 µm above (top) and 2 µm below (bottom) the BC.

### Determination of the BC length fraction

BC length fraction was determined from merged confocal x-y mid-sections of 40× images (25–35 cells); cell–cell-contacting domains, identified by β-catenin or F-actin labeling, and ZO1-labeled BC were traced with the freehand line tool in ImageJ, individual segments were registered in the ROI manager and the summed-up length of all BC in an image was divided by the summed-up length of cell-cell contacting domains. Quantification was done unblinded to the experimental condition.

### Determination of apparent BC width

BC width was determined from each x-y-z maximum-projected ZO1 image; the distance across the five widest BC regions was recorded.

### Cell area determination

Cell area was calculated as the total cell area traced from x-y-z maximum-projected 40× images divided by the number of nuclei in the image (in bi-nucleated cells, only one nucleus was counted).

### BC-associated F-actin area fraction

BC-associated F-actin area fraction was assessed from 40× images of confocal x-y-z stacks of phalloidin-labeled cells; 2 µm projections of the mid sections covering all F-actin-labeled BCs were generated and the area encompassing thresholded BC was determined. The same threshold was then applied to determine the F-actin area in projections of the entire cell volume, including F-actin at top and bottom; presented is the ratio of both measures.

## Data presentation and statistical analysis

Except for graphs in Fig. 4A, where data points are means from five image fields investigated in one experiment, all other data points are derived from individual independent experiments. They each represent averages from quantifications of 3–5 random images encompassing a total of 100–150 cells. For Figs S2A and S5B, five measurements in each of five images per experiment were conducted. Analysis was carried out using GraphPad Prism. Error bars are represented as mean±s.e.m. (Fig. 4A) or mean±s.d. (all others). In Fig. 4A, samples were compared by nonparametric Mann–Whitney test. For all other comparisons, which involved more than two samples one-way ANOVA with Tukey's post-hoc multiple comparisons were applied. We used the convention for representation of $P$-values as follows: *$P<0.05$; **$P<0.01$, ***$P<0.001$.

## Acknowledgements

We are grateful to Fadi-Luc Jaber for performing rat liver perfusions at the beginning of the project.

## Competing interests
The authors declare no competing or financial interests.

## Author contributions
Conceptualization: A.M.; Data curation: A.M., D.C., F.L.-D.; Formal analysis: A.M., D.C., F.L.-D.; Funding acquisition: A.M.; Investigation: D.C., F.L.-D.; Methodology: D.C., F.L.-D.; Project administration: A.M.; Supervision: A.M.; Validation: D.C.; Visualization: A.M., D.C., F.L.-D.; Writing – original draft: A.M.; Writing – review & editing: D.C., F.L.-D.

## Funding
The work was supported by the National Institute of Diabetes and Digestive and Kidney Diseases (NIDDK) (2R01DK118015) and by funds from Albert Einstein College of Medicine to A.M. Open Access funding provided by Albert Einstein College of Medicine. Deposited in PMC for immediate release.

## Data and resource availability
All relevant data and details of resources can be found within the article and its supplementary information.

## Peer review history
The peer review history is available online at https://journals.biologists.com/jcs/lookup/doi/10.1242/jcs.264412.reviewer-comments.pdf

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
