## [Peer Review File · Journal of Cell Science]

Bile canaliculi formation in primary hepatocytes requires $\alpha 1\beta 1$ integrin-dependent adherens junction re-organization

David Cohen, Francisco Lázaro-Diéguez and Anne MÜsch

DOI: 10.1242/jcs.264412

Editor: Kathleen Green

Review timeline

Original submission:	29 August 2025
Editorial decision:	30 September 2025
First revision received:	7 October 2025
Accepted:	11 October 2025

Original submission

First decision letter

MS ID#: jcs.264412

MS TITLE: Bile Canaliculi formation in Primary Hepatocytes requires $\alpha 1\beta 1$ integrin-dependent Adherens Junction Re-organization

AUTHORS: Anne Muesch; David Cohen; Francisco Lazaro-Dieiguez

ARTICLE TYPE: Research Article

Dear Dr Muesch,

We have now reached a decision on the above manuscript.

To see the reviewers' reports and a copy of this decision letter, please go to:

As you will see, the reviewers gave favourable reports but raised some critical points that will require amendments to your manuscript. I hope that you will be able to carry these out because I would like to be able to accept your paper, depending on further comments from reviewers.

Reviewer 1

This manuscript explores bile canalicular formation in primary hepatocytes cultured between two layers of extra cellular matrix. This "sandwich" culture system mimics the environment of hepatocytes in situ flanked by fenestrated endothelial cells on either side of two basal domains. Using confocal imaging, the authors explore how differences in the rigidity of the overlaying vs. plating matrix impacts the establishment of hepatic polarity with an emphasis on adhesions mediated by $\alpha 1\beta 1$ and αV integrins. In general, the imaging is effective and phenotypes apparent. With that being said, the major problem with the manuscript was its overall organization and figure presentation. This was a very difficult read - matching text with images shown took effort and in some cases, I wasn't fully certain what was shown in each panel. This uncertainty made it such that the data presented could not be fully interpreted without tremendous effort, and in some cases, not fully interpreted at all.

In Figure 1, it is not fully clear if all the upper panels in C, D and E are all from cells grown with no overlaying matrix. The small schematics above D and E make it seem like they are meant to

represent all 6 panels below. But do they? It is not fully clear what is being shown in Fig. 1B - and it was only briefly mentioned in the text. Also when discussing Fig. 1, the text jumps to Fig. 2A, then back to Fig. 1, then back to Fig. 2B and C. Fig. 2D isn't described until after getting all the way through Fig. 3. Should it be moved? Perhaps grouping figures differently would help with the flow.

In Fig. 3E, the last two panels (+ α 5B1 panels) aren't described until the details of Fig. 6 are being discussed.

Also, there lacked consistency in figure labeling and order. It would be really helpful if the images and graphs were all arranged with increasing matrix rigidity rather than arbitrarily as is done now. Go from collagen gel to 0.5 kPa, 20 kPa then glass.

Also at times, the label says glass and at others it says collagen-glass. Are those the same condition? Are they all collagen gel on top of glass?

It might also be useful if the authors include a schematic of the distributions of the various integrins, adhesion complex proteins, actin binding proteins, tight junctional components on soft vs. rigid matrix as cells polarize to better summarize the many moving parts presented in the paper.

Reviewer 2

SUMMARY OF THE ADVANCE MADE IN THIS PAPER AND ITS POTENTIAL SIGNIFICANCE TO THE FIELD

This manuscript reports on a study aimed at unraveling the role of specific hepatocyte-ECM interactions in the establishment and maintenance of the unique polarity phenotype, i.e., the formation of a branching canalicular network, of hepatocytes. The authors report the importance of ECM rigidity and implicate two integrin types in this process. Overall, experiments have been performed well with appropriate controls and use of multiple angles of approach where possible. The work advances the field of hepatocyte polarity from a cell biological but less so from a hepatological perspective.

SUGGESTIONS TO AUTHORS

Major comments [Please request additional experiments only if they are essential for supporting the conclusions; authors should be encouraged to highlight any claims that are preliminary or speculative, or to discuss any pitfalls or alternative interpretations in a 'Limitations' section.]

1. Some of the claims made are speculative and should be either 1) rephrased to more accurately fit the experimental evidence or 2) supported by additional experimental evidence. For example, on page 6 (first page of the Results section; line 27), the authors state that "apical AJs were attached to F-actin fibers." However, physical attachment cannot be claimed based on immunofluorescence microscopy experiments. The term "colocalized with" seems more appropriate. The same holds true for the sentence "...and can be seen thugging on ZO-1-lined BC..." Thugging is a dynamic process that cannot be inferred from microscopy images and requires tension measurements. Same again for e.g., "favors" (page 7, line 30), "pulled into" (page 8, line 10), "bear the tension" (page 12, line 22), "recruit" (page 12, line 42), and "contractility" (figure 7 title). Inferring dynamic processes from static microscopy images is highly speculative. The impact of this study would have benefited from additional in-cell tension measurements, for which several methods have been developed.

2. The nonsymmetric ECM rigidity (and the presumed therewith associated forces) in this in vitro overlay model (acknowledged on page 8 (line 43), page 9 (line 46), and page 14 (lines 22-28)) has been instrumental in unraveling the roles of different integrins in the formation of the bile canaliculi but probably does not resemble the in vivo situation. There has been some remark on uneven ECM composition in fibrotic liver (which needs a reference); a more thorough discussion on how the findings in this in vitro model may or may not reflect the in vivo situation is needed.

Minor comments

1. The terms "Young's modulus" and "storage modulus" should be explained for the non-expert reader. The term "pliant" was not known to me, and a more accessible synonym may be recommended.

First revision

Author response to reviewers' comments

On behalf of my co-authors, I want to thank you and the reviewers for your efforts evaluating our manuscript. We are pleased that the reviewers found our study interesting, and we are grateful for the suggestions they made to improve the manuscript.

We have implemented them in the revised version of text and figures; all changes in the text are in red font. Below, we detail these changes point by point.

In response to Reviewer 1: This manuscript explores bile canalicular formation in primary hepatocytes cultured between two layers of extra cellular matrix. This "sandwich" culture system mimics the environment of hepatocytes in situ flanked by fenestrated endothelial cells on either side of two basal domains. Using confocal imaging, the authors explore how differences in the rigidity of the overlaying vs. plating matrix impacts the establishment of hepatic polarity with an emphasis on adhesions mediated by $\alpha 1\beta 1$ and αV integrins. In general, the imaging is effective and phenotypes apparent. With that being said, the major problem with the manuscript was its overall organization and figure presentation. This was a very difficult read - matching text with images shown took effort and in some cases, I wasn't fully certain what was shown in each panel. This uncertainty made it such that the data presented could not be fully interpreted without tremendous effort, and in some cases, not fully interpreted at all.

In Figure 1, it is not fully clear if all the upper panels in C, D and E are all from cells grown with no overlaying matrix. The small schematics above D and E make it seem like they are meant to represent all 6 panels below. But do they? It is not fully clear what is being shown in Fig. 1B - and it was only briefly mentioned in the text. Also when discussing Fig. 1, the text jumps to Fig. 2A, then back to Fig. 1, then back to Fig. 2B and C. Fig. 2D isn't described until after getting all the way through Fig. 3. Should it be moved? Perhaps grouping figures differently would help with the flow.

We have re-organized the data in Figures 1 and 2 to fix these problems. Specifically, we 1) modified the cartoon illustrations to match the experimental conditions they depict with the corresponding images; 2) we deleted former Fig.1B, which was redundant with the message in images in the new Fig.s 2A and 3G; 3) we re-arranged former Fig.2A to Fig.1D to match the order of the figure panels with the order in which they are discussed in the text; 4) we removed the Obtustatin panel from Fig. 2D and moved it to Fig.3H.

In addition, we have moved up in the text the paragraph in which we discuss the $\alpha 1\beta 1$ localization depicted in Fig.3A to be better connected to the rest of the discussion of this figure.

In Fig. 3E, the last two panels ($+\alpha 5\beta 1$ panels) aren't described until the details of Fig. 6 are being discussed.

We removed the $\alpha 5\beta 1$ panels from 3E and added one of the panels to Fig. 6B where $\alpha 5\beta 1$ is discussed

Also, there lacked consistency in figure labeling and order. It would be really helpful if the images and graphs were all arranged with increasing matrix rigidity rather than arbitrarily as is done now. Go from collagen gel to 0.5 kPa, 20 kPa then glass.

We have now uniformly arranged data in graphs with different matrices such that the soft matrix

comes before rigid matrix. However, we have paired gel with glass and 0.5kPa PAA with 20kPa PAA, rather than adopted the suggested “collagen gel to 0.5 kPa, 20 kPa then glass” order the reviewer suggested. We did so because PAA- based matrices and collagen-only matrices are more directly comparable.

Also at times, the label says glass and at others it says collagen-glass. Are those the same condition? Are they all collagen gel on top of glass?

We unified the labeling, leaving out the “coll-I” addition, so that now the label for all samples plated on collagen-coated glass says “glass” and for samples on collagen-gel, says “gel”.

It might also be useful if the authors include a schematic of the distributions of the various integrins, adhesion complex proteins, actin binding proteins, tight junctional components on soft vs. rigid matrix as cells polarize to better summarize the many moving parts presented in the paper.

We added a summary schematic to Fig.7 depicting the localization/organization of integrins and cell-cell junctions on soft and rigid matrix.

In response to Reviewer 2: SUMMARY OF THE ADVANCE MADE IN THIS PAPER AND ITS POTENTIAL SIGNIFICANCE TO THE FIELD

This manuscript reports on a study aimed at unraveling the role of specific hepatocyte-ECM interactions in the establishment and maintenance of the unique polarity phenotype, i.e., the formation of a branching canalicular network, of hepatocytes. The authors report the importance of ECM rigidity and implicate two integrin types in this process. Overall, experiments have been performed well with appropriate controls and use of multiple angles of approach where possible. The work advances the field of hepatocyte polarity from a cell biological but less so from a hepatological perspective.

SUGGESTIONS TO AUTHORS

Major comments [Please request additional experiments only if they are essential for supporting the conclusions; authors should be encouraged to highlight any claims that are preliminary or speculative, or to discuss any pitfalls or alternative interpretations in a 'Limitations' section.]

1. Some of the claims made are speculative and should be either 1) rephrased to more accurately fit the experimental evidence or 2) supported by additional experimental evidence. For example, on page 6 (first page of the Results section; line 27), the authors state that “apical AJs were attached to F-actin fibers.” However, physical attachment cannot be claimed based on immunofluorescence microscopy experiments. The term “colocalized with” seems more appropriate. The same holds true for the sentence “...and can be seen thugging on ZO-1-lined BC...” Thugging is a dynamic process that cannot be inferred from microscopy images and requires tension measurements. Same again for e.g., “favors” (page 7, line 30), “pulled into” (page 8, line 10), “bear the tension” (page 12, line 22), “recruit” (page 12, line 42), and “contractility” (figure 7 title). Inferring dynamic processes from static microscopy images is highly speculative. The impact of this study would have benefited from additional in-cell tension measurements, for which several methods have been developed.

We agree with the reviewer that directly measuring cell-matrix and cell-cell adhesion tension would allow us to make firmer statements regarding the forces at play. We have utilized in Figs 6 and 7 colocalization with Vinculin as a proxy for tension; embarking on direct tension measurements would require the generation and validation of tension sensor expressing adenoviruses, the only means of efficient recombinant gene expression in hepatocytes. We have therefore, as suggested by the reviewer, rephrased the statements that represented an over-interpretation of the data:

“apical AJs were attached to F-actin fibers” was changed to “localized to F-actin fibers”

““tugging” on ZO-1-lined BC” was changed to “contacting ZO1-lined BC”

“Thus, in hepatocyte culture, low matrix rigidity and lack of stress fibers favors the reorganization from substrate-parallel to substrate-perpendicular AJ and TJs, which is associated with faster and more extensive BC formation” reads now: “Thus, in hepatocyte culture, low matrix rigidity and lack

of stress fibers is associated with faster and more extensive BC formation”.

“which, via their associated TJs, were apparently pulled into the cell-cell junctions on top” was altered to “F- actin connected BC remnants, identified by their ZO1 labeling, to the cell-cell junctions on top (Fig.3C)”

Since Vinculin recruitment to matrix or cell-cell adhesions only occurs when these junctions are stretched and Vinculin recruitment is recognized as a proxy for tension, we believe it is appropriate to interpret the colocalization of p120 catenin with vinculin on top as evidence for tension on these junctions. We have nevertheless changed the statement from: “indicating that AJs rather than the fibrillar matrix adhesions likely bear the tension generated by the substrate-parallel stress fibers” to “indicating that AJs but not the fibrillar matrix adhesions are under tension which is likely generated by association of AJs with substrate-parallel stress fibers”.

“Fig. 7: a1b1 and an aV integrin independently regulate AJ contractility at overlay domain and rigid plating surface, respectively” was changed to “: a1b1 and an aV integrin independently regulate AJ- linked substrate-parallel F-actin bundles at overlay domain and rigid plating surface, respectively”.

2. The nonsymmetric ECM rigidity (and the presumed therewith associated forces) in this in vitro overlay model (acknowledged on page 8 (line 43), page 9 (line 46), and page 14 (lines 22-28)) has been instrumental in unraveling the roles of different integrins in the formation of the bile canaliculi but probably does not resemble the in vivo situation. There has been some remark on uneven ECM composition in fibrotic liver (which needs a reference); a more thorough discussion on how the findings in this in vitro model may or may not reflect the in vivo situation is needed.

We have now expanded on the paragraph discussing rigidity versus asymmetry of rigidity and included references. It now reads:

“Our culture model thus cannot isolate effects resulting from top-bottom asymmetry in matrix rigidity from the effects of rigidity *per se*. It is conceivable however that uneven ECM deposition in the space of Disse during development of fibrosis leads to similar asymmetries *in vivo* as well. This possibility is supported by the acknowledgement that due to the focal nature of hepatic stellate cell activation, the main source of fibrogenic collagen in the Space of Disse, collagen deposition is highly heterogenous (Kawamura et al., 2022; Ratziu et al., 2005). Recent 3D volumetric imaging of collagen in liver biopsies confirmed a high labeling variability within the volume of ~200 μm^3 optically cleared liver sections (Yigit et al., 2025), although it still remains to be determined if this variability is present at the level of hepatocyte cords”.

Minor comments

1. The terms “Young’s modulus” and “storage modulus” should be explained for the non-expert reader. The term “pliant” was not known to me, and a more accessible synonym may be recommended.

We have added the explanation for the stiffness measurement and replaced pliant with soft.

Second decision letter

MS ID#: jcs.264412R1

MS Title: Bile Canaliculi formation in Primary Hepatocytes requires a1b1 integrin-dependent Adherens Junction Re-organization

Authors: Anne Muesch; David Cohen; Francisco Lazaro-Dieguez

Article Type: Research Article

Dear Dr Muesch,

I am happy to tell you that your manuscript has been accepted for publication in Journal of Cell Science, pending standard publication integrity checks.